# PrefixKV: Adaptive Prefix KV Cache is What Vision Instruction-Following Models Need for Efficient Generation

**Ao Wang**[1][*]  **Hui Chen**[2][*]  **Jianchao Tan**[3]  **Kefeng Zhang**[3]  **Xunliang Cai**[3]
**Zijia Lin**[1]  **Jungong Han**[4]  **Guiguang Ding**[1][,†]
[1]School of Software, Tsinghua University    [2]BNRist, Tsinghua University
[3]Meituan Inc.    [4]Department of Automation, Tsinghua University

## Abstract

Recently, large vision-language models (LVLMs) have rapidly gained popularity for their strong generation and reasoning capabilities given diverse multimodal inputs. However, these models incur significant computational and memory overhead during inference, which greatly hinders the efficient deployment in practical scenarios. The extensive key-value (KV) cache, necessitated by the lengthy input and output sequences, notably contributes to the high inference cost. Based on this, recent works have investigated ways to reduce the KV cache size for higher efficiency. Although effective, they generally overlook the distinct importance distributions of KV vectors across layers and maintain the same cache size for each layer during the next token prediction. This results in the significant contextual information loss for certain layers, leading to notable performance decline. To address this, we present PrefixKV, where "Prefix" means the top-ranked KV based on importance rather than position in the original sequence. It reframes the challenge of determining KV cache sizes for all layers into the task of searching for the optimal global prefix configuration. With an adaptive layer-wise KV retention recipe based on binary search, the maximum contextual information can thus be preserved in each layer, facilitating the generation. Extensive experiments demonstrate that our method achieves the state-of-the-art performance compared with others. It exhibits superior inference efficiency and generation quality trade-offs, showing promising potential for practical applications. Code is available at https://github.com/THU-MIG/PrefixKV.

## 1    Introduction

Recent years have witnessed the significant advancements of large vision-language models (LVLMs) [37, 66, 25, 35, 2, 13, 10]. Based on the powerful Large Language Models (LLMs) [16, 49, 53, 1, 52, 27, 3, 61], these models integrate visual inputs, showing strong generation and reasoning abilities for various multimodal tasks. They demonstrate inspiring application potential in various fields like autonomous driving [12, 56] and intelligent medical analyses [29, 34].

However, despite their remarkable capabilities, the efficient deployment of LVLMs encounters notable challenges in real-world scenarios. This stems from the typical Transformer architecture employed in the LVLMs, which necessitates the global interaction of tokens. During autoregressive decoding, the key and value vectors of previous tokens are thus required to be stored as the KV cache and subsequently retrieved by the output token [44]. The KV cache grows linearly with the number of

---

[*]Equal contributions. † Corresponding author.

39th Conference on Neural Information Processing Systems (NeurIPS 2025).

processed tokens, which leads to notable memory overhead and heavy burden on GPU communication under lengthy sequences. This causes suboptimal efficiency, resulting in the inference bottleneck.

Given this, recent works have investigated pruning unimportant KV vectors or merging adjacent vectors to reduce the KV cache size while preserving the model performance [40, 68, 60, 67]. For example, $H_2O$ [68] discards less important ones based on the attention scores. Elastic Cache [40] identifies the important KV vectors as the anchor points and merges the surrounding less important cache with these anchors. While effective, existing works [40, 68, 60] typically apply a uniform strategy that retains the same number of KV vectors for each layer to generate next token efficiently, often overlooking the layer-wise heterogeneous characteristics. Our analyses in Sec. 3.2 reveal the notably distinct importance distributions of KV vectors across layers, highlighting the need for tailored retention recipe for each layer adaptively. Intuitively, the importance distribution is concentrated in certain layers while relatively dispersed in other layers. As a suboptimal solution, retaining the same cache size for each layer results in obvious information loss in dispersed layers, and redundant cache in concentrated layers, as shown in Fig. 1.(a).

To address this, we present PrefixKV, a new KV cache compression method for efficient and accurate model generation. We define the prefix[2] of KV cache as the top ones in the priority sequences that are sorted KV vectors according to the normalized importance. The prefix KV vectors of each layer represents the retained cache, making the challenge of determining the optimal layer-wise cache size for the next token generation equivalent to identifying the optimal global prefix configuration. To derive such configuration for a given compression ratio budget, we leverage the prefix cumulative priority as the measurement for the amount of reserved contextual information in each layer. Binary search is then utilized to obtain the desirable information retention ratio, enabling the layer-wise KV retention aligns with the overall budget and maintains the ideal cumulative priority. This ensures that each layer can preserve maximal contextual information after compression, keeping the high

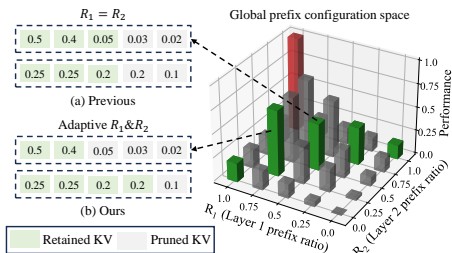

Figure 1: Comparison between previous methods and ours. Previous methods often keep the same prefix length for priority sequences of KV, *i.e.*, retraining the same cache size for each layer. This causes notable information loss for certain layers. In this example, the first layer loses 30% of information. In contrast, we derive the optimal global prefix configuration to preserve as much information as possible in each layer. In this example, both layers can retain 90% of information, thereby enhancing performance.

generation quality of models, as shown in Fig. 1.(b). Extensive experiments show that our method achieves the state-of-the-art performance compared with existing works. It can greatly boost inference efficiency while well maintaining the model's strong capabilities, exhibiting promising potential for real-world applications. Notably, with compression budget of 20%, it provides $1.8\times$ inference speedup for LLaVA-1.5-7B, attaining competitive performance compared with original model.

## 2 Related Work

### 2.1 Vision instruction-following model.

The progress of large vision-language models (LVLMs) has significantly expanded the capabilities of large language models (LLMs) [1, 52, 53, 16, 27, 2, 50, 51, 49] by incorporating visual information, leading to powerful generation and reasoning ability for multimodal tasks [37, 35, 13, 71, 14, 9, 45, 30, 5]. These models typical use linear projection [37] or perceivers [26] to integrate visual representations into the input of LLMs directly. Then, they are further finetuned on high-quality instructional datasets, which include the image-text pairs and the language instructional commands. These models can thus follow multimodal instructions effectively and accurately. Inspired by the notable advancements, subsequent works have sought to enhance their abilities of region-level grounding [7, 66, 43, 10], video comprehension [65, 31], industrial anomaly recognition [22, 62, 8], and biomedical image understanding [29, 54], *etc*.

---

[2]Prefix refers to top elements sorted by importance instead of position.

## 2.2 KV cache compression.

While powerful, existing LVLMs typically encounter high inference costs due to the large parameter count and complex computation, hindering the practical applications. Given this, numerous efforts have focused on improving the inference efficiency, including quantization [33, 59], distillation [21, 47], and pruning [41, 48, 24], *etc*. These methods typically focus on the compression on the model level, reducing inference overhead brought by the parameters. Meanwhile, the KV cache on the data level also incurs significant memory usage and GPU communication overhead during inference. To address this, recent works have investigated various ways to compress the KV cache [60, 40, 23, 20, 46, 55, 57]. For example, StreamingLLM [60] leverages the distance to the output token as the importance indicator and preserves starting KV vectors and those adjacent to the output token. $H_2O$ [68] utilizes the attention scores as importance metric for KV vectors and retains the important ones during generation. KVMerger [57] identifies sets of suitable KV states for merging and uses gaussian kernel weighted aggregation. Elastic Cache [40] divides the KV cache into several buckets according the positions of important KV cache and merges the vectors in the same bucket into one. While effective, they often overlook the distinct importance distribution across layers and retain the same KV cache size in each layer, causing notable information loss. Moreover, despite works like [67, 20, 28] explore layer-wise KV cache size allocation, they often depend on manually designed patterns, which still achieve inferior model's generation capability.

## 3 Methodology

In Sec. 3.1, we first introduce the inference process of LVLMs and the general KV cache compression framework. Then, in Sec. 3.2, we formalize the process of KV compression as retaining the prefix of KV cache and uncover the fact of diverse importance distributions of KV vectors in layers. In Sec. 3.3, we further present PrefixKV, to deliver the ideal layer-wise cache size for the next token prediction by binary searching for the optimal global prefix configuration.

## 3.1 Preliminary

LVLMs exhibit exceptional capabilities for multimodal instructions. Given an input which often consists of the system prompt and user instruction, they first embed the text into the token embeddings through the pre-defined vocabulary and transform the image as flattened patches through the visual encoder. Then, the model enters the prefilling stage, where all tokens interact with each other through the self-attention mechanism. Meanwhile, the key and value vectors of each token are stored as the KV cache, which keeps the contextual information for generation. Subsequently, the model transitions to the decoding stage and outputs the response in an autoregressive way. In each step, the latest predicted token serves as the input and it interacts with the cached KV vectors for preceding information by the self-attention module. It is worth noting that the KV cache size is proportional to the length of processed tokens. This can consume notable memory resources and result in bottleneck in inference speed. It calls for KV cache compression methods to reduce the KV cache size effectively.

The general KV cache compression framework in existing works consist of two stages [40, 68, 60] [3]. Firstly, after prefilling, the importance of each KV vector is derived based on the attention scores or the distance to the generated output. Then, the most important ones are retained in the cache while the less important KV vectors are removed to reduce the memory footprint. Meanwhile, the reserved KV cache sizes in layers meet the requirement of the compression ratio budget. Secondly, during decoding stage, with the inclusion of KV vectors in cache for newly generated tokens, the importance metric of each KV vector is updated. Less critical ones are removed to ensure that the cache size consistently aligns with the overall budget. Our method also follows this general framework, as shown in Fig. 3. Specifically, after prefilling, we retain the most important KV vectors by the ideal global prefix configuration. In the decoding, we follow [40] to prune the vectors at a fixed distance to the latest generated token and maintain the global prefix configuration to meet target budget. Additionally, our method is also complementary to prefilling acceleration ways like FastV [11].

---

[3]For more preliminary, please refer to previous works [44, 69, 63, 70].

## 3.2 Layer-wise KV Cache Importance Distribution

**Mathematical notations for KV compression.** We first introduce the necessary notations to formalize the compression procedure. Specifically, we follow [40, 68] to employ the attention scores as the importance metric, which indicates the amount of contextual information of each KV vector. We suppose that the model consists of $L$ transformer layers. For the $l$-th, layer, its input tokens $\{t_l^1, t_l^2, ..., t_l^N\}$ interact with each other in the multi-head self-attention module, where $N$ is the token number. For the $i$-th head, we denote the query, key, and value vectors of the token $t_l^n$ as $q_l^{i,n}$, $k_l^{i,n}$, and $v_l^{i,n}$, respectively. Then, the causal attention score matrix $A_l^i = \{a_l^{i,m,n}\}_{N \times N}$ can be derived by $a_l^{i,m,n} = \frac{\exp(q_l^{i,m} \cdot k_l^{i,n})}{\sum_{j \le m} \exp(q_l^{i,m} \cdot k_l^{i,j})}$, where $a_l^{i,m,n}$ indicates the attention score of token $t_l^m$ with respect to token $t_l^n$ in the $i$-th head. Then, the total attention score that each token $t_l^n$ receives in the $i$-th head can be derived by $\sum_m a_l^{i,m,n}$. For each $k_l^{i,n}$ and $v_l^{i,n}$, we follow [40] to define their importance metric as the averaged total attention score of $t_l^n$ across all heads, *i.e.*, $I_l^n = \text{Average}_i(\sum_m a_l^{i,m,n})$, where $\text{Average}_i$ denotes the average operation for heads. It is noted that the importance metric of each KV vector varies at each layer but is the same across heads. Then, for a compression ratio budget $r$, the top $R_l$ proportion of KV vectors with the highest importance are retained in each head at the $l$-th layer. Besides, the adoption of $\{R_1, R_2, ..., R_L\}$ satisfy the requirement of $\sum_{l=1}^L R_l N = rLN$.

**Distinct importance distributions across layers.** Existing KV cache compression methods [68, 60, 40] often retain the same number of KV vectors for each layer. This, however, overlooks the diverse contextual information distribution across layers and causes notable valuable information loss. To uncover this fact, we leverage the lorenz curve [19, 18] to characterize the importance distribution across different layers. Specifically, for the $n$-th KV vector of $l$-th layer, we first obtain its importance ratio relative to all tokens by normalization, *i.e.*, $\mathcal{I}_l^n = \frac{I_l^n}{\sum_{j=1}^N I_l^j}$. We then sort the importance set $\{\mathcal{I}_l^1, \mathcal{I}_l^2, ..., \mathcal{I}_l^N\}$ in the descending order to derive the priority sequence of KV vectors. In the priority sequence, the vectors ranked higher have larger importance, making the KV compression equivalent to retaining the prefix of KV cache. Suppose that the sorted indices are $\{s_l^1, s_l^2, ..., s_l^N\}$, for each prefix size ratio $o$ of this sequence, we can obtain its cumulative priority by $P_l^o = \sum_{j \le oN} \mathcal{I}_l^{s_l^j}$. $P_l^o$ indicates the amount of contextual information kept in the $l$-th layer after retaining top $o$ proportion of the most important KV vectors. By deriving the prefix size ratio sequence $\{\frac{1}{N}, \frac{2}{N}, ..., 1\}$ and its corresponding cumulative priority sequence

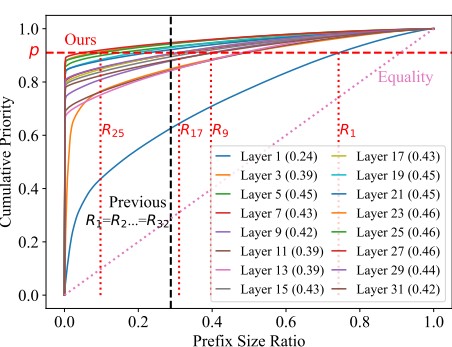

Figure 2: The lorenz curve of priority sequence for KV vectors in different layers. We observe that different layers exhibit diverse importance distributions in the KV cache. Previous methods (the dashed black line) that keep the same prefix cause the notable information loss in layers with dispersed distributions. In contrast, our method (the dashed red line) maximally retains the amount of contextual information of each layer by adaptively maintaining the maximal prefix cumulative priority. The numbers in parentheses in the legend represent the gini coefficient of priority sequence in each layer. A higher gini index indicates a more concentrated importance distribution. It quantitatively demonstrates the varying importance distributions of KV vectors across layers.

$\{P_l^{\frac{1}{N}}, P_l^{\frac{2}{N}}, ..., P_l^1\}$, we can then obtain the lorenz curve of importance distribution. As shown in Fig. 2, we observe that *the cumulative priority growth trends vary significantly across layers*. Previous works generally retain the same layer-wise cache size, *i.e.*, adopting the same prefix size ratio of KV cache with $R_j = r, \forall j \in [1, L]$. In this scenario, as shown in the dashed black line, different layers exhibit markedly distinct cumulative priorities, *i.e.*, $P_l^{R_j}$. This suggests an uneven retention of contextual information across layers, where layers showing rapid growth trends retain a substantial amount, whereas those with slower growth trends retain relatively little. This disparity leads to obvious information loss in layers with slow growth and adversely impacts generation quality. We also quantitatively show the diverse importance distribution by gini coefficient [15, 19]. It is defined as the area that lies between the equality line (the dotted pink line in Fig. 2) and the lorenz curve. The smaller it is, the more uniform the importance distribution, and vice versa. As shown in the legend in

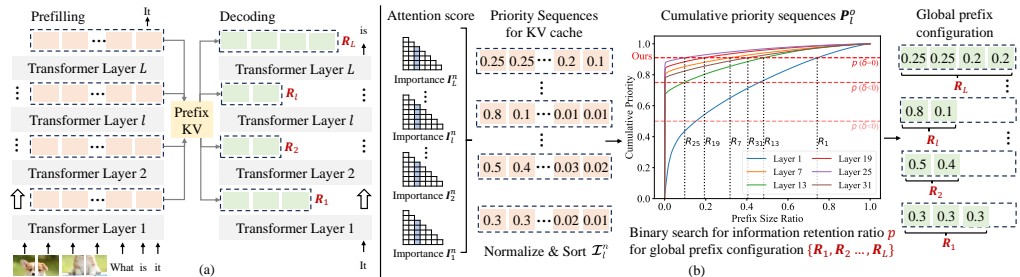

Figure 3: (a) The inference process of LVLMs, where the orange and green rectangles denote the KV cache generated during prefilling and utilized during decoding, respectively. After prefilling, the KV cache is layer-wisely compressed according to the proportions specified by PrefixKV, *i.e.*, $\{\boldsymbol{R}_1, ..., \boldsymbol{R}_L\}$. During decoding, as the sequence lengthens and cache increases, the KV cache consistently maintains the derived compression proportions by pruning KV at a fixed distance [40]. (b) The overview of PrefixKV. It employs binary search for cumulative priority sequences of KV to derive the optimal global prefix configuration, which delivers ideal cache size ratio for each layer.

Fig. 2, different layers exhibit varying gini coefficients, further demonstrating the heterogeneity of KV cache in layers. This calls for adaptively determining the prefix of KV cache for each layer.

### 3.3  PrefixKV

Based on above observations, we introduce PrefixKV. With layer-wise cache size scheme formalized as global prefix configuration, it employs binary search to derive ideal solution, as shown in Fig. 3.(b).

**Global prefix configuration.** All layers' cache size ratios $\{\boldsymbol{R}_1, ..., \boldsymbol{R}_L\}$, *i.e.*, the prefix size ratios of KV cache, constitute the global prefix configuration space of the model. The goal is to identify the optimal global prefix configuration to maintain the high quality of model generation. Given the compression ratio budget $r$, we discover a information retention ratio $p$ to derive such configuration. Specifically, with the priority sequences, $p$ represents the cumulative priority threshold. For the $l$-th layer, its proportion of retained KV vectors is thus the minimum prefix size ratio $o$ such that the cumulative priority $\boldsymbol{P}_l^o$ is larger than or equal to $p$, *i.e.*, $\boldsymbol{R}_l = \min(\{o|\boldsymbol{P}_l^o \geq p\})$. Meanwhile, the value of $p$ satisfies the requirement of the compression ratio budget, *i.e.*,

$$\sum_l \boldsymbol{R}_l = \sum_l \min(\{o|\boldsymbol{P}_l^o \geq p\}) = rL. \tag{1}$$

The corresponding prefix configurations $\{\boldsymbol{R}_1, ..., \boldsymbol{R}_L\}$ ensure that maximal prefix cumulative priority is kept layer-wisely. Finding $p$ is needed for global prefix configuration.

**Binary search for optimal configuration.** Due to the numerous possible values of $p$, obtaining $p$ is quite challenging. We propose employing binary search to efficiently derive $p$ for adaptive layer-wise KV retention. Formally, we start with the interval of $[p_1, p_2]$ where $p_1 = 0$ and $p_2 = 1$. We try $p = \frac{p_1+p_2}{2} = 0.5$ and calculate its corresponding compression budget difference $\delta = \sum_l \boldsymbol{R}_l - rL$. If $\delta$ is equal to zero, we have found the value for $p$. If it is smaller than zero, we set $p_1 = p$; otherwise, we set $p_2 = p$. Then, we update $p = \frac{p_1+p_2}{2}$ and repeat the process until meeting the constraint. Algo. 1 presents the process. In this way, as

---

**Algorithm 1:** Binary Search for ratio $p$

Initialize $p_1 \leftarrow 0$, $p_2 \leftarrow 1$;
**while** $p_1 < p_2$ **do**
  $p \leftarrow \frac{p_1+p_2}{2}$,
  $\sum_l \boldsymbol{R}_l = \sum_l \min(\{o|\boldsymbol{P}_l^o \geq p\})$;
  $\delta = \sum_l \boldsymbol{R}_l - rL$;
  **if** $\delta == 0$ **then return** $p$ ;
  **else if** $\delta < 0$ **then** $p_1 \leftarrow p$ ;
  **else** $p_2 \leftarrow p$ ;
**return** $p$;

---

shown in the dashed red line in Fig. 2, different layers can consistently keep the maximal amount of the contextual information. It ensures that the valuable KV vectors are cached and accessible during generation across all layers. The quality of the model's outputs can be maintained even after significant compression of the KV cache. In practice, $p$ that satisfies $\delta = 0$ may not exist. We can thus set a small threshold for $\delta$ to terminate the search and scale the resulting global prefix configuration to meet the target budget.

Moreover, we observe that the cumulative priority sequences of layers are similar and robust across different samples. Therefore, given a compression ratio budget, we can leverage random samples to

Table 1: Comparison with SOTA methods on LLaVA-Description with the PPL / ROUGE metrics under various compression ratio budgets. Note that a lower PPL is better, while a higher ROUGE is better. The results without the KV compression are 3.20 / 0.62 for LLaVA-1.5-7B and 2.73 / 0.63 for LLaVA-1.5-13B, respectively. It shows that our method consistently achieves superior performance.

| Model | Method | 10% | 20% | 30% | 40% | 50% | 60% | 70% | 80% | 90% |
|---|---|---|---|---|---|---|---|---|---|---|
| 7B | Local | 66.0 / 0.22 | 105 / 0.14 | 70.0 / 0.18 | 47.5 / 0.17 | 33.8 / 0.19 | 14.7 / 0.30 | 5.50 / 0.41 | 4.78 / 0.50 | 4.03 / 0.55 |
| | $H_2O$ | 54.5 / 0.28 | 48.3 / 0.31 | 32.0 / 0.33 | 18.3 / 0.32 | 12.9 / 0.34 | 7.50 / 0.41 | 4.28 / 0.51 | 4.16 / 0.53 | 3.72 / 0.57 |
| | Pyramid | 14.3 / 0.31 | 12.4 / 0.31 | 7.16 / 0.31 | 5.75 / 0.37 | 3.80 / 0.51 | 3.47 / 0.55 | 3.41 / 0.59 | 3.20 / 0.73 | 3.20 / 0.74 |
| | Elastic | 18.0 / 0.29 | 14.0 / 0.29 | 11.8 / 0.29 | 7.38 / 0.32 | 6.31 / 0.36 | 5.97 / 0.39 | 3.66 / 0.54 | 3.55 / 0.55 | 3.58 / 0.57 |
| | Ours | **4.41 / 0.43** | **3.69 / 0.51** | **3.48 / 0.55** | **3.41 / 0.57** | **3.41 / 0.58** | **3.41 / 0.59** | **3.25 / 0.63** | **3.20 / 0.74** | **3.20 / 0.76** |
| 13B | Local | 60.0 / 0.15 | 139 / 0.12 | 56.3 / 0.21 | 16.1 / 0.27 | 13.2 / 0.31 | 7.06 / 0.37 | 3.72 / 0.48 | 3.72 / 0.52 | 3.25 / 0.55 |
| | $H_2O$ | 12.4 / 0.39 | 10.4 / 0.39 | 8.50 / 0.40 | 4.56 / 0.46 | 3.78 / 0.49 | 3.58 / 0.49 | 3.16 / 0.55 | 3.28 / 0.57 | 3.06 / 0.59 |
| | Elastic | 14.9 / 0.30 | 5.75 / 0.35 | 4.41 / 0.40 | 3.55 / 0.50 | 3.36 / 0.52 | 3.28 / 0.53 | 2.97 / 0.58 | 2.89 / 0.60 | 3.02 / 0.59 |
| | Ours | **3.72 / 0.48** | **3.17 / 0.53** | **2.97 / 0.59** | **2.92 / 0.60** | **2.89 / 0.60** | **2.84 / 0.61** | **2.77 / 0.69** | **2.73 / 0.74** | **2.73 / 0.79** |

Table 2: Comparison with SOTA methods on MM-Vet with the PPL / ROUGE metrics under various compression ratio budgets. The results without the KV compression are 5.28 / 0.58 for LLaVA-1.5-7B and 4.72 / 0.58 for LLaVA-1.5-13B, respectively.

| Model | Method | 10% | 20% | 30% | 40% | 50% | 60% | 70% | 80% | 90% |
|---|---|---|---|---|---|---|---|---|---|---|
| 7B | Local | 109 / 0.11 | 90.0 / 0.08 | 99.0 / 0.13 | 99.0 / 0.16 | 66.0 / 0.16 | 28.4 / 0.27 | 12.4 / 0.34 | 7.88 / 0.41 | 6.28 / 0.46 |
| | $H_2O$ | 158 / 0.25 | 120 / 0.26 | 72.5 / 0.29 | 35.3 / 0.31 | 18.6 / 0.30 | 10.3 / 0.39 | 7.09 / 0.44 | 6.22 / 0.46 | 5.72 / 0.49 |
| | Pyramid | 20.8 / 0.26 | 10.4 / 0.28 | 7.50 / 0.30 | 5.75 / 0.34 | 5.63 / 0.46 | 5.50 / 0.46 | 5.41 / 0.53 | 5.28 / 0.73 | 5.28 / 0.75 |
| | Elastic | 40.5 / 0.25 | 21.0 / 0.25 | 14.9 / 0.29 | 11.3 / 0.29 | 9.06 / 0.32 | 7.63 / 0.38 | 5.97 / 0.46 | 5.56 / 0.48 | 5.53 / 0.54 |
| | Ours | **7.38 / 0.39** | **5.97 / 0.41** | **5.72 / 0.46** | **5.53 / 0.46** | **5.50 / 0.48** | **5.44 / 0.50** | **5.38 / 0.59** | **5.28 / 0.74** | **5.28 / 0.77** |
| 13B | Local | 135 / 0.15 | 120 / 0.14 | 77.0 / 0.24 | 53.8 / 0.26 | 40.5 / 0.27 | 18.0 / 0.34 | 9.06 / 0.42 | 6.63 / 0.39 | 5.41 / 0.43 |
| | $H_2O$ | 31.6 / 0.36 | 30.6 / 0.38 | 20.8 / 0.40 | 10.6 / 0.43 | 7.75 / 0.39 | 6.28 / 0.44 | 5.63 / 0.46 | 5.25 / 0.47 | 4.88 / 0.56 |
| | Elastic | 34.3 / 0.28 | 11.6 / 0.34 | 8.00 / 0.37 | 6.31 / 0.44 | 5.81 / 0.44 | 5.44 / 0.49 | 4.97 / 0.52 | 4.81 / 0.51 | 4.81 / 0.56 |
| | Ours | **6.28 / 0.40** | **5.16 / 0.46** | **4.88 / 0.52** | **4.78 / 0.52** | **4.72 / 0.55** | **4.72 / 0.57** | **4.72 / 0.64** | **4.69 / 0.75** | **4.72 / 0.79** |

derive the corresponding $p$ value and the optimal global prefix configuration $\{R_1, R_2, ..., R_L\}$ for each layer offline, as analyzed in Fig. 4, Tab. 6, and Tab. 5. The configuration can thus be adopted for the model during inference, which avoids the online binary search and shows good generalizability.

# 4 Experiments

## 4.1 Experimental Settings

We follow [40] to employ LLaVA-1.5-7B [35] and LLaVA-1.5-13B [35], and leverage the LLaVA-Description [40] and MM-Vet [64] instruction-following datasets for evaluation. LLaVA-Description is a curated subset of 1000 detailed description instructions from the LLaVA-1.5 training set [40]. MM-Vet encompasses a diverse set of tasks designed to comprehensively evaluate the model performance in both understanding and generation. Besides, we also re-conduct the instruction tuning for LLaVA-1.5 models, to exclude the LLaVA-Description for preventing data leakage during evaluation. We follow [40] to employ the perplexity (PPL) and the ROUGE score [32] metrics. Specifically, PPL quantifies the exponential value of the cross-entropy loss between the predicted next token and ground truth. A lower PPL indicates the better generation quality. The ROUGE score calculates the longest common subsequence between the generated output and the reference outputs, with F1 score used for evaluation. A higher ROUGE score indicates the better consistency with reference responses.

Following [40], we utilize the model without cache compression to generate the reference output for the ROUGE score evaluation. Besides, in practice, multiple generation texts of a model can be different when the temperature is not zero. Thus, for the compression budget of 100%, *i.e.*, without compression, we measure the ROUGE score of two generation outputs and it is thus smaller than 1. For other budgets, we use the temperature of 0 for the reproducibility and the ROUGE score may thus exceed the uncompressed one. We simply leverage 10 random samples from the training set of model for the global prefix configuration estimation offline. During decoding, we follow [40] to fix the distance to the latest token for pruning and maintain the layer-wise cache size ratios to satisfy the target budget. We compare with the state-of-the-art Elastic Cache [40], Heavy-Hitter Oracle [68],

Table 3: Inference time for our method under compression ratio budgets of 20% / 40% / 60% / 80%. OOM indicates out of memory.

| Batch Size | Model Size | Token Length | Latency (s) | | Throughput (token/s) | |
|---|---|---|---|---|---|---|
| | | | PrefixKV | Full Cache | PrefixKV | Full Cache |
| 8 | 13B | 1024+512 | 20.0 / 24.3 / 27.5 / 29.7 | 30.5 | 204.6 / 168.1 / 148.7 / 137.6 | 134.1 |
| 16 | 13B | 624+256 | 11.7 / 14.2 / 15.9 / 17.3 | 17.8 | 349.5 / 288.0 / 256.3 / 236.5 | 230.2 |
| 16 | 7B | 1024+512 | 16.8 / 22.5 / 26.6 / 29.5 | 30.7 | 486.7 / 363.3 / 307.9 / 276.9 | 266.6 |
| 48 | 7B | 624+256 | 13.1 / OOM / OOM / OOM | OOM | 934.4 / OOM / OOM / OOM | OOM |

StreamingLLM [60], and PyramidKV [67], which are termed as Elastic, H$_2$O, Local, and Pyramid for brevity, respectively.

## 4.2 Main Results

As shown in Tab. 1 and Tab. 2, our method consistently achieves the state-of-the-art performance compared with others across various compression budgets and different model scales. For example, as shown in Tab. 1, under a compression budget of 50% on LLaVA-Description, our PrefixKV significantly outperforms H$_2$O and Elastic cache by 9.49 and 2.90 in PPL for LLaVA-1.5-7B. It also surpasses the Local cache by a notable margin of 0.39 in ROUGE score. We also observe that our advantages over others can be further amplified as the compression budget decreases. Under a compression budget of 20%, our PrefixKV can still maintain a satisfactory PPL of 3.69, which is reduced by 8.71, 44.6, and 10.3 compared with Pyramid, H$_2$O, and Elastic, respectively. For large model of LLaVA-1.5-13B, our method also demonstrates the notable superiority over others. For example, as shown in Tab. 2, our PrefixKV outperforms Elastic and H$_2$O cache by 1.53 and 5.82 PPL, respectively, under a compression budget of 40% on MM-Vet. Furthermore, our method shows the minimal performance decline compared with the original model without KV compression in various scenarios. For example, as shown in Tab. 2, for LLaVA-1.5-13B, our method can maintain the nearly identical PPL and ROUGE scores compared with the uncompressed model at any compression budget exceeding 30%. Under a compression budget of 20%, our PrefixKV only leads to the decline of 0.44 PPL. These results well demonstrate the superiority of ours.

## 4.3 Model Analyses

We present comprehensive analyses for our method. Following [40], experiments are conducted on MM-Vet based on LLaVA-1.5-7B with PPL for evaluation, by default, .

**Inference efficiency.** We evaluate the inference efficiency of the model with our KV compression method to verify its benefit for acceleration. We follow [40] to conduct the evaluation in two practical scenarios. Specifically, firstly, we construct the input with 1024 prompt tokens and generate 512 tokens. Secondly, we employ a shorter input with 624 prompt tokens and generate 256 tokens, which means the minimal prompt length with only the image tokens and system prompts. We measure the inference time on the NVIDIA A100 GPU and use the batch size which maximizes the available memory to simulate the realistic deployment scenarios for better efficiency and throughput. As shown in Tab. 3, our method shows the notable inference speedups under various compression budgets compared with original model with full KV cache. For example, under the compression budget of 20% with the batch size of 16 for LLaVA-1.5-7B, our method demonstrates 1.8× inference speedup in terms of throughput. Besides, it also reduces the memory usage, avoiding OOM and enabling efficient inference with large batch size of 48. We also note that in this scenario, our method maintains the superior performance with only 0.69 PPL decline compared with the original model. These results well demonstrate the benefit of our method in practical deployment for efficient LVLMs.

**Global prefix configuration matters.** We verify the effectiveness of our method in identifying the ideal global prefix configuration. We first introduce the baseline, which keeps the same retained KV cache size for all layers. As shown in Tab. 4, our PrefixKV consistently brings performance benefit under various compression budgets. For example, under the compression budget of 30%, it surpasses the baseline by significant margin of 14.7 PPL. Besides, as the compression budget gradually decreases from 90% to 10%, its superiority becomes increasingly evident, showing increasing

Table 4: Prefix config. (Uncompressed: 5.28).

| Method | 10% | 20% | 30% | 40% | 50% | 60% | 70% | 80% | 90% |
|---|---|---|---|---|---|---|---|---|---|
| Baseline | 41.8 | 26.6 | 20.4 | 15.4 | 11.8 | 9.06 | 6.47 | 5.75 | 5.72 |
| Pyramid | 20.8 | 10.4 | 7.50 | 5.75 | 5.63 | 5.50 | 5.41 | **5.28** | **5.28** |
| PrefixKV | **7.38** | **5.97** | **5.72** | **5.53** | **5.50** | **5.44** | **5.38** | **5.28** | **5.28** |

Table 5: Sample numbers. (Uncompressed: 5.28).

| Number | 10% | 20% | 30% | 40% | 50% | 60% | 70% | 80% | 90% |
|---|---|---|---|---|---|---|---|---|---|
| 1 | 7.63 | 6.03 | 5.72 | 5.53 | 5.53 | 5.44 | 5.38 | 5.28 | 5.28 |
| 5 | 7.50 | 6.03 | 5.72 | 5.53 | 5.50 | 5.41 | 5.38 | 5.28 | 5.28 |
| 10 | 7.38 | 5.97 | 5.72 | 5.53 | 5.50 | 5.44 | 5.38 | 5.28 | 5.28 |
| 20 | 7.38 | 5.97 | 5.72 | 5.53 | 5.50 | 5.41 | 5.38 | 5.28 | 5.28 |

performance improvements. We also compare ours with PyramidKV [67], which manually allocates larger cache size in shallow layers and smaller size in deep layers. Since it compresses the cache only after the prefilling, we integrate it in the baseline for fair comparisons. As shown in Tab. 4, our strategy achieves superior performance over PyramidKV in various scenarios, especially under the low compression budget. These results show that compared with the same cache size across layers and the manually designed scheme by PyramidKV, our method can identify better global prefix configuration, which retains the overall contextual information more effectively.

**Effect of offline estimation.** Given a compression budget, we estimate the optimal KV cache size for each layer by random samples offline, which avoids the overhead of online searching. To verify its effectiveness, we first examine the variation of retained KV cache size ratios at each layer across different samples. As shown in Fig. 4, different samples exhibit the similar cache size ratios for each layer. This indicates the potential of using the samples to estimate the optimal cache size ratios offline. We further present the comparison results between offline estimation and online binary searching for each sample in Tab. 6. It can be observed that offline estimation can obtain the comparable performance with the online searching. Therefore, our method can be integrated into the models efficiently, without the extra inference overhead. We also inspect the impact of adopting different numbers of random samples. As shown in

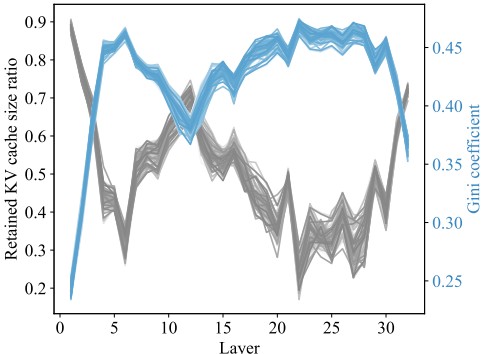

Figure 4: The retained KV cache size ratios for each layer of 100 random samples under the compression ratio of 50% and their gini coefficients of the priority sequences for KV vectors in each layer. It shows that different samples exhibit similar and robust characteristics, showing the reasonableness of offline estimation.

Tab. 5, the performance is stable and robust across different sample sizes. Besides, employing a single sample can achieve good performance, which shows the effectiveness of offline estimation.

**Robustness of offline estimation.** To inspect the robustness of offline estimation, we measure the standard deviation (std) of the retained KV cache size ratios obtained via online binary search across all samples, as well as the mean absolute difference (mad) between these values and the configuration of offline estimation. We present the results in Tab. 7. It quantitatively shows that the variance of retained KV cache size ratios across samples is small, indicating the feasibility of offline estimation. The small mean absolute difference supports the efficacy of offline estimation. We also conduct experiments using random samples from different domains and with different instruction complexities to further verify the robustness. Specifically, for different domains, we leverage random samples from open-knowledge visual question answering (VQA), optical character recognition (OCR), and complex reasoning (reasoning), respectively. In Tab. 8, we present the PPL results. We observe that the performance remains stable across different domains. Besides, for instruction complexities, we obtain random samples with different instruction lengths: shorter than 100, between 100 and 500, and longer than 500. We present the results in Tab. 9. It shows that our method also exhibits robustness to the instruction complexity, showing favorable generalizability. For the binary search for offline estimation, we also present analyses for its sensitivity and convergence behavior. Specifically, we conduct experiments under the compression ratio of 50% using varying approximation threshold $\delta$ to inspect the robustness of binary search. We present the average search steps and PPL results in Tab. 10. It shows that the search consistently converges within 7-12 steps, with stable performance across different settings. We also fix the number of search steps of binary search for evaluation. As shown in Tab. 11, the performance remains stable across difference scenarios, showing the reliability.

Table 6: Offline estimation.(Uncompressed: 5.28).

| Method | 10% | 20% | 30% | 40% | 50% | 60% | 70% | 80% | 90% |
|---|---|---|---|---|---|---|---|---|---|
| Offline | 7.38 | 5.97 | 5.72 | 5.53 | 5.50 | 5.44 | 5.38 | 5.28 | 5.28 |
| Online | 7.38 | 5.97 | 5.66 | 5.53 | 5.50 | 5.41 | 5.38 | 5.28 | 5.28 |

Table 7: Deviation analyses.

| | 10% | 20% | 30% | 40% | 50% | 60% | 70% | 80% | 90% |
|---|---|---|---|---|---|---|---|---|---|
| std | 0.010 | 0.021 | 0.023 | 0.024 | 0.023 | 0.020 | 0.013 | 0.011 | 0.006 |
| mad | 0.010 | 0.021 | 0.024 | 0.024 | 0.023 | 0.020 | 0.014 | 0.011 | 0.006 |

Table 8: Robustness to domain.

| Domain | 10% | 20% | 30% | 40% | 50% | 60% | 70% | 80% | 90% |
|---|---|---|---|---|---|---|---|---|---|
| VQA | 7.45 | 5.97 | 5.72 | 5.53 | 5.50 | 5.41 | 5.38 | 5.28 | 5.28 |
| OCR | 7.38 | 6.01 | 5.72 | 5.53 | 5.53 | 5.41 | 5.38 | 5.28 | 5.28 |
| reasoning | 7.40 | 5.97 | 5.72 | 5.53 | 5.50 | 5.44 | 5.38 | 5.28 | 5.28 |

Table 9: Robustness to instruction complexity.

| Length | 10% | 20% | 30% | 40% | 50% | 60% | 70% | 80% | 90% |
|---|---|---|---|---|---|---|---|---|---|
| (0, 100] | 7.38 | 6.01 | 5.66 | 5.53 | 5.53 | 5.44 | 5.38 | 5.28 | 5.28 |
| (100, 500] | 7.42 | 5.97 | 5.72 | 5.53 | 5.50 | 5.44 | 5.38 | 5.28 | 5.28 |
| (500, inf) | 7.38 | 5.97 | 5.72 | 5.54 | 5.50 | 5.41 | 5.38 | 5.28 | 5.28 |

Table 10: Robustness to $\delta$

| $\delta$ | 0.0125 | 0.025 | 0.05 | 0.075 | 0.1 |
|---|---|---|---|---|---|
| avg. step | 11.5 | 10.8 | 9.3 | 8.5 | 7.8 |
| PPL | 5.50 | 5.50 | 5.50 | 5.51 | 5.53 |

Table 11: Robust. to step

| steps | 7 | 8 | 9 | 10 | 11 | 12 |
|---|---|---|---|---|---|---|
| PPL | 5.53 | 5.51 | 5.51 | 5.50 | 5.50 | 5.50 |

Table 12: Merging. (Uncompressed: 5.28).

| Method | 10% | 20% | 30% | 40% | 50% | 60% | 70% | 80% | 90% |
|---|---|---|---|---|---|---|---|---|---|
| PrefixKV | 7.38 | 5.97 | 5.72 | 5.53 | 5.50 | 5.44 | 5.38 | 5.28 | 5.28 |
| Position | 7.63 | 6.06 | 5.75 | 5.53 | 5.44 | 5.31 | 5.31 | 5.28 | 5.28 |
| Feature | 7.38 | 5.97 | 5.72 | 5.53 | 5.44 | 5.31 | 5.28 | 5.28 | 5.28 |

**Relation between KV cache size and importance distribution.** We derive better KV cache size for each layer based on the importance distribution. To provide deeper insights for their relation, we visualize the KV cache size ratios and the gini coefficients of priority sequence for different samples in Fig. 4. We observe that layers with higher gini coefficients, *i.e.*, with more concentrated importance distributions typically have fewer KV cache sizes, and vice versa. This qualitatively validates the reliability of our method, as layers with concentrated importance distributions can retain most information with fewer KV vectors, while layers with more dispersed importance distributions require larger KV cache sizes. Besides, we note that the retained KV cache size exhibits a W-shaped trend across layers. This inspires that utilizing more cost-effective attention mechanism in shallow to mid layers, as well as in mid to deep layers, may lead to more efficient model architecture. Besides, we also observe that the distributions of retained KV cache size ratios and Gini coefficients obtained from different samples exhibit low variance, indicating favorable consistency in importance patterns. The empirical stability further verifies the reliability of offline estimation using few samples, as they can approximate the overall layer-wise importance distribution.

**Eviction or merging.** Our PrefixKV shows favorable performance by cache eviction, *i.e.*, retaining the prefix vectors and pruning the unimportant ones. To mitigate the information loss of eviction, previous research explores merging the less important KV vector with its nearest important one in position [40]. Therefore, we inspect the performance variation under combining our PrefixKV with cache merging. In addition to merging based on positional distance, we also examine the way based on the feature similarity, which are explored in the token merging field [6, 58]. Specifically, we denote the $l$-th layer's retained KV vector set as $\Omega_l^r$ and pruned set as $\Omega_l^p$, respectively. The cache merging operation is the same for key and value vectors in different heads, we thus proceed with the key vector for illustration and omit the head superscript. For each pruned key vector $\boldsymbol{k}_l^m$ where $m \in \Omega_l^p$, we obtain its matching metric $\boldsymbol{c}_l^{mn}$ to each retained one $\boldsymbol{k}_l^n$ where $n \in \Omega_l^r$. For each $\boldsymbol{k}_l^n$, we obtain the set of pruned vectors that match with it by $\boldsymbol{T}_l^n = \{m \in \Omega_l^p | n = \mathrm{argmax}_u(\boldsymbol{c}_l^{mu})\}$. The vectors are then merged to update $\boldsymbol{k}_l^n$ by $\boldsymbol{k}_l^{n*} = \mathrm{Average}(\{\boldsymbol{k}_l^u | u \in \boldsymbol{T}_l^n \cup \{n\}\})$. We experiment with matching metrics based on the positional distance and feature cosine similarity by $\boldsymbol{c}_l^{mn} = -|m - n|$ and $\boldsymbol{c}_l^{mn} = \frac{\boldsymbol{k}_l^m \cdot \boldsymbol{k}_l^n}{||\boldsymbol{k}_l^m|| ||\boldsymbol{k}_l^n||}$, which are denoted as "Position" and "Feature", respectively. As shown in Tab. 12, integrating "Position" can lead to inferior performance over PrefixKV in certain scenarios. We note that our method enables the maximal preservation of important KV vectors, and merging based on the positional distance, however, could introduce the interference to important cache due to the feature discrepancies among the vectors [58]. Besides, we observe that with less feature dispersion, combining "Feature" can bring the marginal improvements. This indicates that our method can well retain the significant contextual information across layers and eliminate the need for cache merging to reduce information loss, demonstrating its efficacy.

**Analyses for the feature disturbance.** To further verify the effectiveness of our method in preserving valuable contextual information across layers, we inspect the feature perturbation caused

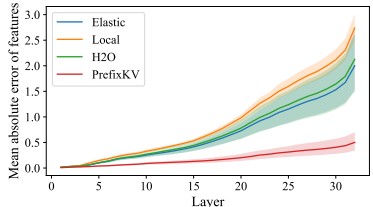

Figure 5: Mean absolute error vis.

Table 13: Results on Qwen-VL. (Uncompressed: 6.28).

| Method | 10% | 20% | 30% | 40% | 50% | 60% | 70% | 80% | 90% |
|---|---|---|---|---|---|---|---|---|---|
| Local | 314 | 185 | 93.0 | 54.5 | 70.0 | 43.8 | 24.6 | 17.5 | 11.4 |
| $H_2O$ | 72.5 | 58.0 | 43.3 | 29.3 | 19.1 | 15.4 | 12.9 | 11.1 | 9.50 |
| Elastic | 404 | 66.0 | 33.8 | 20.4 | 12.8 | 10.4 | 8.63 | 8.63 | 8.63 |
| Ours | **16.6** | **9.94** | **8.50** | **8.13** | **7.88** | **7.09** | **6.41** | **6.28** | **6.28** |

by KV cache compression for output tokens at each layer. Specifically, for each output token, we calculate the mean absolute error between its feature with and without compression. We employ the compression budget of 50% and visual the average error across output tokens for 100 random samples across layers. As shown in Fig. 5, our method consistently exhibits lower mean absolute error between features compared with others. This shows that our method can introduce less interference to the features of output tokens, demonstrating enhanced retention of contextual information and resulting in improved generation quality.

**Generalizability on other LVLMs.** Following [40], we conduct experiments on Qwen-VL [2] to verify the general effectiveness of our method. In Tab. 13, our method consistently exhibits superiority over others across various compression budgets, highlighting generalizability.

**Comparison results on LLMs.** Our method can also be adopted for the KV cache compression in LLMs. Following [67], we conduct experiments on LongBench [4] and Needle-in-a-HayStack [38, 17] based

Table 14: Results on LLMs.

| Dataset | Base. | Local | $H_2O$ | Pyramid | Ours |
|---|---|---|---|---|---|
| LongB. | 41.6 | 32.0 | 35.0 | 37.1 | **37.5** |
| Needle. | 100 | 30.4 | 49.3 | 97.3 | **97.6** |

Table 15: With quant.

| Method | 12.5% | 25% | 50% | 100% |
|---|---|---|---|---|
| KIVI | 21.8 | 41.2 | 41.6 | 41.6 |
| Ours | 40.2 | 41.1 | 41.5 | 41.6 |

on LLaMA-3-8B [16]. We set the KV cache size budget to 128 and present the results in Tab. 14. We can see that our PrefixKV also outperforms others in such scenarios. It well reserves the valuable information for model's generation in limited KV cache budget but under long context. The results further exhibit the general efficacy of our method. Besides, we also compare ours with the KV cache quantization methods for LLMs. We present the comparison results with advanced KIVI [39] on LongBench for LLaMA-3-8B in Tab. 15. We find that in long context scenarios, for different budgets, KV cache favors different compression dimensions, *i.e.*, precision in quantization and length in pruning. For extreme 12.5% compression budget, the length dimension shows more redundancy and our method performs better. For larger budgets, removing precision redundancy preserves more information and KIVI obtains slightly better performance. Moreover, our pruning method is complementary to quantization method and can be combined to eliminate redundancy in both dimensions. For example, combining KIVI under 4 bit with ours can achieve competitive performance of 40.0 with 3.125% compression ratio budget, leading to stronger efficacy.

# 5 Conclusion

In this paper, we present PrefixKV to effectively compress KV cache for efficient generation of large vision-language models (LVLMs). It derives the optimal KV cache size for each layer by searching for the ideal global prefix configuration with the priority sequences of KV. Maximal preservation of contextual information is thus ensured layer-wisely, contributing to high-quality model generation. Extensive experiments show that our method achieves the state-of-the-art performance compared with others. It provides notable inference speedups while maintaining the generation capability, showing the superiority for practical applications.

# 6 Acknowledgments

This work was supported by National Natural Science Foundation of China (Nos. 62525103, 624B2082, 62271281, 62441235) and Beijing Natural Science Foundation (L247026). It was also supported by Tsinghua University - Meituan Joint Institute for Digital Life. We are also grateful to Jiaxin Li for his thoughtful contribution.

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

Table 16: Multi-round and long text VQA scenarios.

| Dataset | Baseline | Local | $H_2O$ | Elastic. | Ours |
|---|---|---|---|---|---|
| Multi-round | 3.2 | 28.4 | 7.3 | 5.8 | **3.5** |
| Long text | 2.6 | 14.3 | 6.1 | 3.3 | **2.7** |

Table 17: High-resolution scenarios.

| Dataset | Baseline | Local | $H_2O$ | Elastic. | Ours |
|---|---|---|---|---|---|
| MM-Vet | 5.4 | 68068 | 7817 | 4514 | **6.4** |

## A   Implementation Details

We conduct experiments using Pytorch [42] on NVIDIA 3090 GPUs. Following [40], we leverage two different metrics, *i.e.*, PPL and ROUGE score [32], to comprehensively assess the performance of model with KV cache compression.

Specifically, PPL measures the similarity between the probability distribution predicted by the model and that of the ground truth. Given the prefix prompt of $x$, suppose that the sequence of ground-truth subsequent words is $t_1 t_2 ... t_N$, we can then obtain the cross-entropy loss for each predicted probability distribution of vocabulary and ground-truth word by

$$l_i = -\log(p(t_i|x, t_1 t_2 ... t_{i-1})), \tag{2}$$

where $l_i$ denotes the cross-entropy loss for the $i$-th predicted word and $p(t_i|x, t_1 t_2 ... t_{i-1})$ indicates the predicted probability of word $t_i$ under the input sequence of $x$ and $t_1 t_2 ... t_{i-1}$. Then, PPL can be derived by the exponential of the average loss across words, *i.e.*,

$$\text{PPL} = \exp(\frac{\sum_i l_i}{N}). \tag{3}$$

The lower score of PPL represents the closer predicted probability distribution to the ground truth, suggesting stronger generative capabilities of the model.

Moreover, the ROUGE score quantifies the similarity between the model's generated content and the reference sentences. We utilize the widely adopted ROUGE-L F1 score, which identifies the longest common subsequence (LCS) between them. To calculate it, we first obtain the ROUGE-L precision score, which represents the proportion of the LCS found in the model's generation. Besides, the ROUGE-L recall score is assessed by evaluating the proportion of the LCS present in the reference content. The F1 score can then be derived by

$$\text{F1} = \frac{2 \cdot (\text{precision} \cdot \text{recall})}{\text{precision} + \text{recall}}. \tag{4}$$

The higher score of ROUGE F1 represents that the model's output is more similar to the reference content, indicating the better retention of the model's abilities. Besides, when the KV cache compression ratio budget is 100%, we set the temperature to 0.2 to measure the ROUGE score of two generation results, following [40].

## B   More Model Analyses

We present more experiments and analyses for our method in different settings. Specifically, following [24], we evaluate our method in multi-round and long text VQA scenarios. Since the datasets in [24] are not released, we construct them following the implementation details. We experiment with LLaVA-1.5-7B [35] under 50% compression budget and report the PPL (lower is better) in Tab. 16, where "Baseline" indicates the uncompressed result. It can be observed that our method also significantly outperforms others on such benchmarks, further showing its superiority. We also employ LLaVA-NeXT-7B [36] for assessing our method in scenarios with high-resolution images. We conduct experiments on MM-Vet [64] under the compression ratio budget of 50%. Tab. 17 presents the

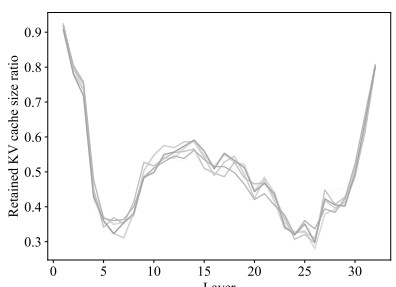

Figure 6: The retained KV cache size ratios of 10 random samples under 50% compression ratio for Qwen-VL.

Table 18: Our method can effectively preserve both the model's overall and fine-grained perceptual abilities. The model adequately captures the overall background, like snowy landscape and clear blue sky, as well as the fine-grained details, like at least ten people skiing. In contrast, with other methods, the model may struggle to provide comprehensive perception.

| User | What's happening in the scene? |
|---|---|
| Local | The snowyards, |
| $H_2O$ | The image of a group of people are skiers are skiing down a snowy mountain slope, the image of a group of people are skiing down a snowy mountain scene of people are skiing down a snowy mountain... |
| Elastic | The image shows a group of people are skiers are enjoying a snowy mountain skiing in the snowy mountain scene with a group of people are skiers are skiing down a snowy mountain. |
| Ours | The image captures a group of people skiing on a snowy mountain slope. There are at least ten people visible in the scene, scattered across the slope, enjoying the winter sport. Some of the skiers are closer to the foreground, while others are further back, creating a sense of depth in the image. The snowy landscape and the clear blue sky make for a beautiful backdrop for the winter sports enthusiasts. |

PPL results (lower is better). It shows that our method also obtains notable improvement compared with others. These results well verify the efficacy and practicality of our method in various scenarios. Besides, we also inspect the retained cache size distribution of Qwen-VL for different samples in Fig. 6. It presents the similar W-shaped trend across layers as in Fig. 4. We hypothesize this similar behavior stemming from the hierarchical processing of LVLMs: (1) Shallow layers captures fine-grained syntactic and visual features, with dispersed foundational token-level information across KV cache vectors and favor larger cache size. (2) Deep layers influence output reasoning and generation. The contextual information helps preserve the fidelity of high-level representations essential for final prediction, relying on larger retention cache. (3) Middle layers serve as intermediate abstraction stage, with feature representations aggregated and compressed. The importance distribution of KV cache vectors is thus more concentrated, requiring fewer cache vectors.

## C  More Chat Generation Results

We present comparative analyses of PrefixKV with different KV cache compression methods [60, 68, 40] in Tab. 18, Tab. 19, Tab. 20, Tab. 21, and Tab. 22. We set the KV cache budget to 20% for inspection based on LLaVA-1.5-7B [35]. It can be observed that in various challenging scenarios, our method can consistently provide comprehensive and helpful responses by retaining the overall contextual information across layers effectively. For example, with our method, the model can reasonably infer that the man carrying a surfboard possibly prepares to go surfing in the water, in Tab. 19. It can also maintain the model's capability of recognizing the six different cakes in Tab. 20 and perceiving the somewhat messy overall atmosphere in Tab. 22. Besides, it can well help the model observe various elements during generation, including the clear blue sky in Tab. 18 and palm trees in Tab. 19, *etc*. In contrast, with other methods, the model struggles to generate coherent outputs, with missing elements, repetitive texts, and abrupt sentences. The results highlight the superiority of our method over others, showing its effectiveness and robustness for practical applications.

Table 19: Our method can retain the valuable contextual information for generation effectively. For example, with our method, the model can make reasonable inference regarding the context of possibly preparing to go surfing for the man with the surfboard.

| User | Can you elaborate on the elements of the picture provided? |
|---|---|
| Local | The man, the beach, the beach, the beach, the beach, the beach, the beach,... |
| H$_2$O | The image features a man is walking along the beach scene shows a man is walking along the beach is a man is walking along the beach... |
| Elastic | The image shows a man is a man is standing on a man is standing on a beach, the beach, the beach, the image features a man is a surfboard. |
| Ours | The image features a man walking along a beach, carrying a surfboard under his arm. He appears to be heading towards the water, possibly preparing to go surfing. The beach is lined with palm trees, adding to the tropical atmosphere. There are several other people scattered around the area, some closer to the water and others further back on the beach. |

Table 20: Our method can help maintain the focus on key elements in the image, like wooden tray and cakes. The model is thus capable of accurately describing the characteristics of salient elements, such six cakes and different shapes. In contrast, with other methods, the model may overlook important elements in the responses.

| User | What is this photo about? |
|---|---|
| Local | The table, the table, the table, the table, the table, the table... |
| H$_2$O | The image features a wooden cake stand is a wooden cake display of a wooden cake display of a cake is a wooden cake is a wooden cake... |
| Elastic | The image of a wooden table is a wooden table is a wooden table is a wooden table... |
| Ours | The image features a wooden tray with a variety of small cakes and pastries on it. There are a total of six cakes on the tray, each with different shapes and sizes. The tray is placed on a wooden surface, and the arrangement of the cakes creates an appealing presentation. |

Table 21: With our method, the model can identify the main features of the image with the low KV cache budget. It can accurately understand the scene and the relationships among salient objects.

| User | Can you describe the main features of this image for me? |
|---|---|
| Local | The Airplane plane plane plane plane plane plane plane? |
| $H_2O$ | The scene of the image shows a large airport scene of a group of people are a group of people are gathered around the airport scene of a group of people... |
| Elastic | The scene of the image of a large airport scene of a group of people are gathered around a crowd of people are gathered around a group of people are a group of people are standing in front of a large airplane, which is a large airplane. The airplane. The airplane is parked on the tarmac. The airplane is parked on the runway. The airplane is parked on the tarmac. The airplane is parked on the tarmac... |
| Ours | The image features a large United States Air Force airplane parked on a runway, with a group of people gathered around it. There are at least 13 people visible in the scene, with some standing closer to the airplane and others further away. The individuals appear to be observing the aircraft or possibly waiting for boarding. The airplane is positioned on the runway, with the main body of the aircraft visible in the foreground and the tail end extending towards the background. |

Table 22: With our method, the model can provide accurate and detailed descriptions. Compared with others, our method can help the model discover various things and perceive the somewhat messy atmosphere after KV cache compression.

| User | What are the key elements in this picture? |
|---|---|
| Local | The books, books on a bookshelves of books on a bookshelf books on a bookshelf books on a bookshelf books... |
| $H_2O$ | The image of a bedroom scene of a bedroom with a bedroom with a bedroom scene of a bedroom with a bedroom... |
| Elastic | The image of a bed is a bed is a bed is a bed is a bed is a bed is a bed is a bed is a bed ... |
| Ours | The image features a bed with a large pile of books on it. The books are of various sizes and are spread across the bed, with some lying horizontally and others vertically. The bed is situated in a room with a window, and the overall atmosphere appears to be somewhat messy. |

# D   Limitation and Societal Impact

**Limitation.** Due to the limited academic resources, we do not evaluate our method on models with larger scales. In addition, the combination of our method with other acceleration ways like distillation and pruning is also worth exploring, which we leave as the future work.

**Societal Impact.** Our proposed method can benefit the widely adoption of large vision-language models in various scenarios, like autonomous driving and medical image analyses, *etc*. We are also aware of the potential risks associated with improper use and committed to preventing this.

