# OpenReview forum: "PrefixKV: Adaptive Prefix KV Cache is What Vision Instruction-Following Models Need for Efficient Generation"
_NeurIPS.cc/2025/Conference — NeurIPS 2025 poster_

### Official Review · Reviewer_ByA6 · 2025-06-30

**Clarity:** 3
**Significance:** 3
**Originality:** 3
**Rating:** 5
**Confidence:** 4

**Summary:**

The paper introduces PrefixKV, an innovative method for KV cache compression in LVLMs. It adjusts layer-wise KV cache retention ratio based on varying importance distribution of KV vectors across layers. By utilizing offline binary search to determine the optimal prefix configuration, PrefixKV aims to enhance inference efficiency while maintaining high-quality generation. The paper presents extensive experiments, showing superiority in terms of inference speed and performance.

**Questions:**

See the weaknesses.

**Ethical Concerns:**

["NO or VERY MINOR ethics concerns only"]

**Final Justification:**

My concerns are well addressed in the rebuttal, and I maintain my original score to accept this paper.

**Limitations:**

Yes

**Quality:**

3

**Strengths And Weaknesses:**

Strengths:

1.It is interesting to leverage the varying layer-wise KV importance for developing a KV cache budget strategy with the analyses from the Lorenz curve and Gini coefficient perspectives. The offline estimation of the global prefix configuration ensures that the method does not incur additional inference costs, making it practical for deployment.
2. The method achieves state-of-the-art performance across various compression ratios, maintaining high performance even at low compression ratios. The experiments are conducted across multiple models and benchmarks, which present comprehensive analyses and comparisons and verify the efficacy of the proposed method.

Weaknesses:
1. The related work section is somewhat short. A more detailed introduction and discussion of recent KV cache compression techniques would provide a clearer context.
2. The "prefix" in PrefixKV might be somewhat misleading. Specifically, "prefix" usually denotes the initial segment of a sequence, but the proposed method preserves tokens based on significance, rather than position in the sequence.

---

> ### Author Rebuttal · Authors · 2025-07-30
>
> We sincerely appreciate your valuable feedback and insightful comments. Thank you for liking the extensive experiments, practical for deployment, and state-of-the-art performance. We provide the response to each comment in the point-to-point manner below. Please feel free to contact us if you have further concerns.
>
> Q1: The related work section is somewhat short. A more detailed introduction and discussion of recent KV cache compression techniques would provide a clearer context.
>
> A1: Thanks. We will enrich the related work section to provide a more comprehensive overview and clearer context in the revision. For example, we will elaborate on the background and advantages of KV cache compression. We will also provide more detailed descriptions of the motivations, methods, and results of KV cache compression techniques, and further enrich the discussions and analyses.
>
> Q2: The "prefix" in PrefixKV might be somewhat misleading. Specifically, "prefix" usually denotes the initial segment of a sequence, but the proposed method preserves tokens based on significance, rather than position in the sequence.
>
> A2: Thanks. We use the term "prefix" to refer to the top KV cache vectors sorted by importance score. We understand the concern and will clarify this more early and clearly in the revision. We will also consider revising the name of our method, e.g., TopKV, to avoid misleading.

---

> > ### Comment · Reviewer_ByA6 · 2025-08-06
> >
> > Thank you for your response. My concerns are well addressed in this rebuttal. As such, I keep my original rate to accept this paper.

---

### Official Review · Reviewer_K11i · 2025-07-02

**Clarity:** 3
**Significance:** 3
**Originality:** 2
**Rating:** 4
**Confidence:** 4

**Summary:**

The paper proposes a method for efficient attention key-value caching in the context of LVMs. In particular, the authors propose a variable-rate caching mechanism based on the relative importance of cached values across layers. For a fixed ``budget” compression rate, the method aims to optimally estimate the best amount of key-value pairs to be cached per layer, with such optimal ratio found through binary search. The authors apply their method across a variety of models including LlaVa and MM-Vet, as well as in Qwen-VL and LLM models. A set of ablation studies show the efficiency of their proposed approach. The proposed method shows to produce state of the art results for high levels of compression, illustrating the benefit of the proposed approach in low-regime scenarios.

**Questions:**

My questions are embedded in the weaknesses section above. In particular:

- Novelty/results tradeoff against PyramidKV

- Offline estimation being on par with online estimation

- Merging tokens better than discarding them.

**Ethical Concerns:**

["NO or VERY MINOR ethics concerns only"]

**Final Justification:**

The rebuttal has partially addressed my concerns and therefore, considering the other reviewers' consideration for the paper, I choose to keep my original rating. I would like to ask the authors to further investigate the reason why offline and online estimation produce similar results, and add such findings to the final manuscript.

**Limitations:**

No, there is not such section in the paper. The authors should add such section in the paper.

**Quality:**

3

**Strengths And Weaknesses:**

Strengths:


The paper proposes a simple yet efficient idea, namely that of varying the number of cached tokens per layer, according to their estimated relevance to future attention maps.


I find it interesting the fact that offline results are on par with online results, illustrating that some pre-computed relevance is enough to have good and efficient compression.


The results show that for high compression levels their proposed approach is better than alternative methods like H2O or Elastic.


The paper includes a good amount of ablation studies including the tradeoff between compression ratio and results, the effect of online vs offline caching, or the effect of merging rather than discarding tokens.


Weaknesses:


The idea, while overall efficient, is relatively simple and the novelty is a bit limited. I see that in Table 4, the proposed approach is almost on par with PyramidKV except for very big compression rates. The comparison novelty-results against PyramidKV is in my opinion a bit limited.



I find it a bit surprising that the Offline estimation produces nearly exactly the same results as that of online caching. This is very surprising as one would expect that online caching should always be related to current sequences and therefore some variability would be expected in these results in favor of online caching. In a similar way, I find it really surprising that a single sample is enough for the ratio optimality search in an offline setting, with results that are pretty much the same as those of online caching. A single example might contain a rather different semantic information and provoke completely different activations in the attention maps at different layers. The results suggest that the percentage of attention values that are relevant is almost fixed for each layer and is independent of the input content. In other words, the same percentage of tokens will be relevant at each layer for a completely variety of samples, regardless the entropy of each input sequence. This requires further investigation or explanation because it is rather surprising.




The results in Table 7 also suggest that merging tokens is a better option than directly disregarding them. This requires further investigation against methods for token merging.

---

> ### Author Rebuttal · Authors · 2025-07-30
>
> We sincerely appreciate your valuable feedback and insightful comments. Thank you for liking the state of the art results, simple yet efficient idea, and a good amount of ablations. We provide the response to each comment in the point-to-point manner below. Please feel free to contact us if you have further concerns.
>
> Q1: The idea, while overall efficient, is relatively simple and the novelty is a bit limited. I see that in Table 4, the proposed approach is almost on par with PyramidKV except for very big compression rates. The comparison novelty-results against PyramidKV is in my opinion a bit limited.
>
> A1: Thanks. Unlike PyramidKV which employs a manually-designed allocation strategy, we quantitatively analyze the inter-layer disparity of KV cache importance distribution across layers through the new cumulative priority sequence perspective, and optimize the global prefix configuration for KV cache retention by binary search. It enables **adaptive** layer-wise cache size budget **with maximal contextual information**. In Table 4, we note that when compression is light, more KV cache redundancy remains in layers, so both methods can preserve sufficient information to maintain generation quality. However, under aggressive compression ratios, the important KV cache vectors are more easily to be discarded and the KV cache size allocation thus becomes more critical. In these scenarios, our method consistently outperforms PyramidKV by notable margins. This underscores the practical advantage and effectiveness of our method, especially in situations where high KV cache compression ratio is essential, such as on resource-constrained mobile and edge devices.
>
> Q2: I find it a bit surprising that the Offline estimation produces nearly exactly the same results as that of online caching. This is very surprising as one would expect that online caching should always be related to current sequences and therefore some variability would be expected in these results in favor of online caching. In a similar way, I find it really surprising that a single sample is enough for the ratio optimality search in an offline setting, with results that are pretty much the same as those of online caching. A single example might contain a rather different semantic information and provoke completely different activations in the attention maps at different layers. The results suggest that the percentage of attention values that are relevant is almost fixed for each layer and is independent of the input content. In other words, the same percentage of tokens will be relevant at each layer for a completely variety of samples, regardless the entropy of each input sequence. This requires further investigation or explanation because it is rather surprising.
>
> A2: Thanks. We understand the concern. We note that as shown in Fig. 4 in the paper and the analyses in A5 for Reviewer rNU4, the distributions of cumulative priority sequence are similar and exhibit beneficial low variance across different samples, domains and instruction complexities. Such property enables offline estimation from a few random samples to provide a favorable approximation of the overall optimal global prefix configuration, i.e., the layer-wise KV cache retention size. In A4 for Reviewer hgee, we also measure the standard deviation (std) of the retained KV cache size ratios obtained via online binary search across all samples, and the mean absolute difference (mad) between those and the configuration of offline estimation. The quantitative results of small standard deviation and small mean absolute difference further verify the effectiveness of offline estimation. The small standard deviation also verifies the feasibility of offline estimation by one sample. We will incorporate these results in the revision and also release the code for the analyses. Although offline estimation may not yield the optimal configuration for every individual input, we hypothesize that it can still identify a generally effective pattern that is less dependent on specific samples and more reflective of the underlying learning mechanisms of model. This hypothesis offers a potentially interesting perspective for understanding the internal learning dynamics of models, which we will further investigate in future work.
>
> Q3: The results in Table 7 also suggest that merging tokens is a better option than directly disregarding them. This requires further investigation against methods for token merging.
>
> A3: Thanks. We acknowledge the better option of merging tokens compared with directly dropping them. Meanwhile, we note that the improvements are relatively marginal. Following your suggestion, we will also explore more effective merging techniques in the future work.
>
> Q4: There is not limitation section in the paper. The authors should add such section in the paper.
>
> A4: Thanks. We present the limitation section in the supplementary.

---

> > ### Comment · Reviewer_K11i · 2025-08-06
> >
> > I appreciate the authors' response. I have read the explanations regarding the offline estimation being on par with online estimation. I believe this requires further exploration yet it does not hinder the current submission.
> >
> >
> > Similarly, I still believe the novelty w.r.t. PyramidKV is not significant, although the proposed approach seems to outperform it in terms of high compression rates and this gives the current work its merit.
> >
> > Therefore I keep my score above the bar.

---

### Official Review · Reviewer_rNU4 · 2025-07-03

**Clarity:** 3
**Significance:** 3
**Originality:** 2
**Rating:** 4
**Confidence:** 3

**Summary:**

This paper tackles the KV cache overhead in large vision-language models (LVLMs). The core argument is that the importance of KV cache vectors is heterogeneous across layers, a claim supported by an analysis of attention distributions. To leverage this, the authors propose PrefixKV, a method that assigns a unique, adaptive compression ratio to each layer. These layer-specific ratios are determined by searching for a single information retention threshold that satisfies a global compression budget. Crucially, this configuration is calculated offline, imposing no overhead during inference.

**Questions:**

(1) A major practical advantage of your method is the offline estimation of the global prefix configuration from a small sample set. Could you provide more analysis on the robustness of this process? For instance, how much does the derived configuration vary when using different small sets of calibration samples, especially samples from different domains or with different instruction complexities? A convincing analysis, even on a small scale, demonstrating the stability of the derived configuration would significantly strengthen the paper's practical claims.
(2) The use of different temperature settings for the baseline (temp > 0) and the proposed method (temp = 0) leads to counter-intuitive ROUGE scores where PrefixKV surpasses the full-cache model (Table 1). Could you please provide results from a revised experiment where both the full-cache baseline and PrefixKV use the exact same deterministic generation settings (i.e., temperature 0)? This would provide a more direct and interpretable measure of information preservation.
(3) The comparison with PyramidKV [65] in Table 4 is crucial, as it isolates the benefit of your search-based adaptive strategy from a manually-designed one. Could you elaborate on your interpretation of these results? Specifically, what is the core mechanism that allows your method to so clearly outperform PyramidKV, especially at aggressive compression ratios (e.g., 10-30%)? A more detailed analysis would help to better situate the novelty of your contribution, moving it beyond "layer-wise adaptation is good" to "our method for finding the adaptation strategy is superior."
(4) The W-shaped curve for retained cache sizes in Figure 4 is an interesting finding. Is there any intuition as to why shallow and deep layers require more cache than middle layers? Is this pattern consistent across different model families, or is it specific to the LLaMA architecture? A brief discussion on this could offer valuable insights into the inner workings of LVLMs and enhance the paper's significance.

**Ethical Concerns:**

["NO or VERY MINOR ethics concerns only"]

**Final Justification:**

The authors responded thoughtfully and strengthened the paper. While minor concerns remain, the work presents a solid contribution to the field and meets the acceptance criteria at this stage.

**Limitations:**

yes

**Quality:**

3

**Strengths And Weaknesses:**

	Strengh:
(1) The paper's approach is well-motivated and principled, grounded in a rigorous analysis of KV cache importance heterogeneity. The use of Lorenz curves provides a strong quantitative justification for the proposed layer-wise compression.
(2) The PrefixKV method is straightforward and elegant. It cleverly reframes a complex multi-layer optimization problem into a simple binary search for a global threshold, and its ability to be pre-computed offline makes the solution highly practical for deployment.
(3) The experiments show that the method surpasses previous methods significantly.

	Weakness:
(1) While the paper presents a well-engineered and effective solution, the core idea of layer-wise KV cache allocation is not entirely novel. The authors acknowledge prior works [65, 19, 27] that explore this direction. The primary contribution thus shifts from the conceptual breakthrough of "adaptive allocation" to the specific algorithm for achieving it (binary searching for a global cumulative priority threshold), which, while elegant, can be seen as a strong incremental improvement rather than a fundamentally new paradigm.
(2) The practicality of the method hinges on the claim that an optimal prefix configuration can be determined offline from a very small number of samples (as few as one, per Table 6). While experimental results are stable, this assumption might not hold under more diverse, out-of-distribution, or adversarial scenarios. The paper lacks a rigorous analysis of the conditions under which this offline estimation remains robust, which is a critical aspect for real-world reliability.
(3) The experimental setup for ROUGE evaluation, where the uncompressed reference is generated with non-zero temperature while compressed models use temperature 0 (lines 237-239), complicates interpretation. This leads to reported ROUGE scores for the proposed method that are higher than the "full cache" baseline (e.g., 0.76 vs 0.62 in Table 1), which is counter-intuitive. A more direct comparison using the same generation parameters for both the baseline and the proposed method would provide a clearer measure of quality preservation.
(4) The most direct competitor, PyramidKV [65]—which employs a non-uniform, "manually designed" layer-wise allocation—is only compared in a supplementary analysis (Table 4). For a stronger claim, PrefixKV should be benchmarked against PyramidKV and other potential adaptive baselines within the main result tables (Tables 1 and 2). The current presentation, which primarily contrasts with uniform allocation methods, may overstate the performance gains attributable to the proposed search strategy versus the general benefit of adaptivity.

---

> ### Author Rebuttal · Authors · 2025-07-30
>
> We sincerely appreciate your valuable feedback and insightful comments.
>
> Q1: While the paper presents a well-engineered...
>
> A1: Thanks. While prior works have explored layer-wise KV cache allocation, our contribution lies in introducing a novel analytical perspective. Instead of relying on manual or heuristic ways, we propose the cumulative priority sequence to capture KV cache importance distribution across layers, using Lorenz curves and Gini coefficients to quantify inter-layer disparity. This reframes the KV cache compression as a global prefix configuration problem. Our binary search method naturally follows from this formulation, offering an efficient and principled solution. We believe this perspective provides fresh insights into KV cache compression for LVLMs.
>
> Q2: The practicality of the method hinges on the claim...
>
> A2: Thanks. We obtain random samples from the diverse training corpus of model that covers various scenarios, including open-knowledge visual question answering, optical character recognition, region-level understanding, detailed description, and complex reasoning, etc. Besides, the random samples for offline estimation are disjoint from the benchmarks, showing the robustness of our method under out-of-distribution scenarios. For adversarial scenarios, which lacks available standardized dataset for LVLMs, we acknowledge it as a valuable direction and will investigate in the future work. Additionally, we conduct experiments to evaluate our method in different practical settings, including multi-round and long text VQA scenarios in Table 1 in the supplementary, and high-resolution scenarios in Table 2 in the supplementary. The results show that our method enjoys favorable performance, verifying the practical reliability.
>
> Q3: The experimental setup for ROUGE evaluation...
>
> & Q6: The use of different temperature settings...
>
> A3 & A6: Thanks. We follow prior work [1] for ROUGE evaluation, where for the compression budget of 100%, the score is measured with non-zero temperature. The baseline results also align with the prior work [1], which leads to counter-intuitive comparison. We understand the concern and conduct experiments for the baseline with the same generation parameters, which results in 0.78 / 0.80 for LLaVA-1.5-7B / 13B in Table 1. We will incorporate these results in the revision.
>
> Q4: The most direct competitor, PyramidKV...
>
> A4: Thanks. We conduct experiments with PyramidKV for comparisons in Table 1 and Table 2 based on LLaVA-1.5-7B. The following tables present the benchmark results. We observe that our method consistently outperforms it across datasets and compression ratios. These results suggest that our method can identify better KV cache size allocation for compression. We think that this stems from our method's global prefix configuration optimization from cumulative priority sequences across layers, allowing layer-wise maximal contextual information preservation and thus better remaining performance even at high compression ratios. Please refer to A7 for more detailed explanation. We will include more comparison results in the revision to better highlight the improvements of our method.
>
> Tab 1. PPL / ROUGE score on LLaVA-Description.
>
> | Method    | 10%         | 20%         | 30%         | 40%         | 50%         | 60%         | 70%         | 80%         | 90%         |
> | --------- | ----------- | ----------- | ----------- | ----------- | ----------- | ----------- | ----------- | ----------- | ----------- |
> | PyramidKV | 14.3 / 0.31 | 12.4 / 0.31 | 7.16 / 0.31 | 5.75 / 0.37 | 3.80 / 0.51 | 3.47 / 0.55 | 3.41 / 0.59 | 3.20 / 0.73 | 3.20 / 0.74 |
> | Ours      | 4.41 / 0.43 | 3.69 / 0.51 | 3.48 / 0.55 | 3.41 / 0.57 | 3.41 / 0.58 | 3.41 / 0.59 | 3.25 / 0.63 | 3.20 / 0.74 | 3.20 / 0.76 |
>
> Tab 2. PPL / ROUGE score on MM-Vet.
>
> | Method    | 10%         | 20%         | 30%         | 40%         | 50%         | 60%         | 70%         | 80%         | 90%         |
> | --------- | ----------- | ----------- | ----------- | ----------- | ----------- | ----------- | ----------- | ----------- | ----------- |
> | PyramidKV | 20.8 / 0.26 | 10.4 / 0.28 | 7.50 / 0.30 | 5.75 / 0.34 | 5.63 / 0.46 | 5.50 / 0.46 | 5.41 / 0.53 | 5.28 / 0.73 | 5.28 / 0.75 |
> | Ours      | 7.38 / 0.39 | 5.97 / 0.41 | 5.72 / 0.46 | 5.53 / 0.46 | 5.50 / 0.48 | 5.44 / 0.50 | 5.38 / 0.59 | 5.28 / 0.74 | 5.28 / 0.77 |
>
> Q5: A major practical advantage of your method is the offline...
>
> A5: Thanks. We conduct experiments using random samples from different domains and with different instruction complexities to verify the robustness of our offline estimation. Specifically, for different domains, we leverage random samples from open-knowledge visual question answering (VQA), optical character recognition (OCR), and complex reasoning (reasoning), respectively. In the below table, we present the PPL results on MM-Vet based on LLaVA-1.5-7B. We observe that the performance remains stable across different domains, showing the robustness.
>
> | domain    | 10%  | 20%  | 30%  | 40%  | 50%  | 60%  | 70%  | 80%  | 90%  |
> | --------- | ---- | ---- | ---- | ---- | ---- | ---- | ---- | ---- | ---- |
> | VQA       | 7.45 | 5.97 | 5.72 | 5.53 | 5.50 | 5.41 | 5.38 | 5.28 | 5.28 |
> | OCR       | 7.38 | 6.01 | 5.72 | 5.53 | 5.53 | 5.41 | 5.38 | 5.28 | 5.28 |
> | reasoning | 7.40 | 5.97 | 5.72 | 5.53 | 5.50 | 5.44 | 5.38 | 5.28 | 5.28 |
>
> Besides, for instruction complexities, we obtain random samples with different instruction lengths: shorter than 100, between 100 and 500, and longer than 500. We present the results in the below table. It shows that our method also exhibits robustness to the instruction complexity, showing favorable generalizability.
>
> | Instruction         | 10%  | 20%  | 30%  | 40%  | 50%  | 60%  | 70%  | 80%  | 90%  |
> | ------------------- | ---- | ---- | ---- | ---- | ---- | ---- | ---- | ---- | ---- |
> | length <= 100       | 7.38 | 6.01 | 5.66 | 5.53 | 5.53 | 5.44 | 5.38 | 5.28 | 5.28 |
> | 100 < length <= 500 | 7.42 | 5.97 | 5.72 | 5.53 | 5.50 | 5.44 | 5.38 | 5.28 | 5.28 |
> | length > 500        | 7.38 | 5.97 | 5.72 | 5.54 | 5.50 | 5.41 | 5.38 | 5.28 | 5.28 |
>
> Additionally, we also follow A4 for Reviewer hgee to measure the mean absolute difference between retained KV cache size ratios obtained via online binary search for all samples and offline-estimated ones by samples with different domains and instruction complexities. The average results across ratios are (VQA: 0.016), (OCR: 0.017), (reasoning: 0.016), (length <= 100: 0.016), (100 < length <= 500: 0.017) , and (length > 500: 0.018). This further shows the favorable robustness of our offline estimation.
>
> Q7: The comparison with PyramidKV in Table 4 is crucial...
>
> A7: Thanks. We think that the core mechanism lies in that our method **globally** optimizes the allocation of KV cache across layers based on cumulative priority sequences. It ensures that each layer retains its most informative portion of KV cache, preserving maximal contextual information. In contrast, PyramidKV employs a manually-designed one (larger cache size in shallow layers and smaller size in deep layers) that may discard high-priority KV cache in some layers while preserving less important ones in others. This allocation becomes especially suboptimal at aggressive compression ratios (e.g., 10-30%), because important KV vectors are more likely to be dropped due to the limited budget. This results in noticeable contextual information loss and clearly inferior performance compared with ours.
>
> Q8: The W-shaped curve for retained cache sizes...
>
> A8: Thanks. We conduct experiments to inspect the retained cache size distribution of Qwen-VL under the compression ratio of 50%, which also presents the similar pattern (Due to that the image is not allowed in the response, we present the average results in the below table).
>
> | Layer     | 1      | 2      | 3      | 4      | 5      | 6      | 7      | 8      | 9      | 10     | 11     | 12     | 13     | 14     | 15     | 16     |
> | --------- | ------ | ------ | ------ | ------ | ------ | ------ | ------ | ------ | ------ | ------ | ------ | ------ | ------ | ------ | ------ | ------ |
> | Ratio     | 0.91   | 0.76   | 0.73   | 0.41   | 0.34   | 0.34   | 0.35   | 0.40   | 0.51   | 0.53   | 0.57   | 0.56   | 0.57   | 0.58   | 0.54   | 0.51   |
> | **Layer** | **17** | **18** | **19** | **20** | **21** | **22** | **23** | **24** | **25** | **26** | **27** | **28** | **29** | **30** | **31** | **32** |
> | Ratio     | 0.54   | 0.52   | 0.50   | 0.43   | 0.47   | 0.43   | 0.35   | 0.33   | 0.37   | 0.31   | 0.40   | 0.37   | 0.43   | 0.50   | 0.63   | 0.80   |
>
> Our intuition for this draws from the hierarchical processing of LVLMs: (1) Shallow layers are responsible for capturing fine-grained syntactic and visual features from the input, where the foundational token-level information is dispersed across KV cache vectors and favor larger retained cache size. (2) Deep layers directly influence output reasoning and generation. The contextual information across KV cache vectors helps preserve the fidelity of high-level representations essential for final prediction, which relies on larger retention cache size. (3) Middle layers often serve as an intermediate abstraction stage, where feature representations are aggregated and compressed. The importance distribution of KV cache vectors is thus more concentrated, which requires fewer cache vectors to be retained. Prior works on pruning and redundancy in models [2,3] also show that middle layers exhibit more representational redundancy, making them more tolerant to compression. We will incorporate more discussion in the revision to offer more insights.
>
> [1] Efficient Inference of Vision Instruction-Following Models with Elastic Cache. ECCV 2024.
>
> [2] Streamlining Redundant Layers to Compress Large Language Models. ICLR 2025.
>
> [3] ShortGPT: Layers in Large Language Models are More Redundant Than You Expect. ACL Findings 2025.

---

### Official Review · Reviewer_hgee · 2025-07-05

**Clarity:** 3
**Significance:** 2
**Originality:** 2
**Rating:** 4
**Confidence:** 3

**Summary:**

This paper proposes PrefixKV, a method for adaptive KV cache compression in large vision-language models (LVLMs), targeting efficient generation during inference. The core idea is to optimize the retained key-value (KV) cache size per layer, rather than applying a uniform retention policy.  PrefixKV introduces a binary search mechanism to derive a global prefix configuration that maximizes contextual information retention while respecting a global compression budget. The method is evaluated on several models (LLaVA-1.5-7B, 13B, Qwen-VL) and datasets (LLaVA-Description, MM-Vet), showing superior performance to state-of-the-art methods such as H2O and Elastic Cache under a range of compression budgets

**Questions:**

1. Could you expand on why 10 samples suffice for offline estimation across different datasets and tasks? Are there scenarios where this assumption might break?
2. How well does a prefix configuration estimated on one dataset (e.g., LLaVA-Description) generalize to another (e.g., MM-Vet)? Have you tested cross-task portability?

**Ethical Concerns:**

["NO or VERY MINOR ethics concerns only"]

**Limitations:**

yes

**Quality:**

2

**Strengths And Weaknesses:**

Strengths.

-  The paper provides strong motivation for layer-wise KV retention, backed by empirical observations such as Lorenz curves and Gini coefficients. The insight into variable importance distributions across layers is well-founded and novel.
- The method provides substantial inference speedups (e.g., 1.8× on LLaVA-1.5-7B) while retaining performance, making it highly relevant for real-world deployment.

Weaknesses.
- While the binary search approach is efficient, the paper does not deeply analyze the sensitivity or convergence behavior of the algorithm (e.g., robustness to approximation thresholds, search steps).
- Some figures (e.g., Fig. 1 and Fig. 3) are packed and slightly difficult to interpret. More concise visuals or splitting into sub-figures could improve clarity.
- The term “prefix” is overloaded in the Transformer literature and may confuse readers. A brief clarification early on (e.g., prefix in sorted importance vs. positional prefix) would help.

---

> ### Author Rebuttal · Authors · 2025-07-30
>
> We sincerely appreciate your valuable feedback and insightful comments. Thank you for liking the strong motivation, well-founded and novel insight, substantial inference speedups, and relevant for real-world deployment. We provide the response to each comment in the point-to-point manner below. Please feel free to contact us if you have further concerns.
>
> Q1: While the binary search approach is efficient, the paper does not deeply analyze the sensitivity or convergence behavior of the algorithm (e.g., robustness to approximation thresholds, search steps).
>
> A1: Thanks. We first conduct experiments under the compression ratio of 50% using varying approximation threshold $\delta$ to evaluate the sensitivity and convergence behavior of binary search. We present the average search steps and PPL results on MM-Vet with LLaVA-1.5-7B in the below table. It shows that the search consistently converges within 7–12 steps, with stable performance across different settings, showing the robustness.
>
> | $\delta$ | Average search step | Result |
> | -------- | ------------------- | ------ |
> | 0.0125   | 11.5                | 5.50   |
> | 0.025    | 10.8                | 5.50   |
> | 0.05     | 9.3                 | 5.50   |
> | 0.075    | 8.5                 | 5.51   |
> | 0.1      | 7.8                 | 5.53   |
>
> Furthermore, we also fix the number of search steps of binary search for evaluation, to verify the robustness of our approach to different search steps. As shown in the below table, the performance remains stable across difference scenarios, showing the reliability. We will include more results and analyses in the revision.
>
> | Search step | Result |
> | ----------- | ------ |
> | 7           | 5.53   |
> | 8           | 5.51   |
> | 9           | 5.51   |
> | 10          | 5.50   |
> | 11          | 5.50   |
> | 12          | 5.50   |
>
> Q2: Some figures (e.g., Fig. 1 and Fig. 3) are packed and slightly difficult to interpret. More concise visuals or splitting into sub-figures could improve clarity.
>
> A2: Thanks. We will split the packed figures, e.g., Fig.1 and Fig.3, into clearer subfigures with more concise visuals to improve readability. For example, for Fig.1, we will organize (a) and (b) into one subfigure, and present the global prefix configuration space into another subfigure. For Fig.3, for better interpretation, we will move the prefilling and decoding processes into one subfigure, and the overview of our method into another subfigure with clearer layout.
>
> Q3: The term “prefix” is overloaded in the Transformer literature and may confuse readers. A brief clarification early on (e.g., prefix in sorted importance vs. positional prefix) would help.
>
> A3: Thanks. To mitigate confusion, in the revision, we will clarify early and clearly that our usage of the term "prefix" refers to the top KV cache vectors based on importance scores, not their original sequence positions. For example, we will add ", where "Prefix" means the top-ranked KV based on importance rather than position in the original sequence" immediately after "we present PrefixKV" in Line 12.
>
> Q4: Could you expand on why 10 samples suffice for offline estimation across different datasets and tasks? Are there scenarios where this assumption might break?
>
> A4: Thanks. We use random samples in training set to estimate the cumulative priority sequences of layers and determine the optimal global prefix configuration offline. As shown in Fig. 4 in the paper, the distributions of retained KV cache size ratios and Gini coefficients obtained from different samples exhibit low variance, indicating favorable consistency in importance patterns. The empirical stability justifies the use of as few as 10 samples for offline estimation, as they can approximate the overall layer-wise importance distribution. We also measure the standard deviation (std) of the retained KV cache size ratios obtained via online binary search across all samples, as well as the mean absolute difference (mad) between these values and the configuration of offline estimation. We present the results in the below table. It quantitatively shows that the variance of retained KV cache size ratios across samples is small, indicating the feasibility of offline estimation. The small mean absolute difference also supports the effectiveness of offline estimation. We will incorporate these results in the revision and also release the code for the analyses.
>
> | Ratio | 10%   | 20%   | 30%   | 40%   | 50%   | 60%   | 70%   | 80%   | 90%   |
> | ----- | ----- | ----- | ----- | ----- | ----- | ----- | ----- | ----- | ----- |
> | std   | 0.010 | 0.021 | 0.023 | 0.024 | 0.023 | 0.020 | 0.013 | 0.011 | 0.006 |
> | mad   | 0.010 | 0.021 | 0.024 | 0.024 | 0.023 | 0.020 | 0.014 | 0.011 | 0.006 |
>
> Besides, we also acknowledge that in special settings (e.g., adversarial scenario), samples may exhibit higher variability in cumulative priority sequences. Consequently, increasing the number of samples may be required to ensure reliable offline estimation. In this way, the overall distribution can be approximated more accurately.
>
> Q5: How well does a prefix configuration estimated on one dataset (e.g., LLaVA-Description) generalize to another (e.g., MM-Vet)? Have you tested cross-task portability?
>
> A5: Thanks. We use random samples from the training set of model for offline estimation of global prefix configuration, which is then leveraged on the evaluation benchmarks under zero-shot settings. Since the training set is disjoint from the evaluation benchmarks, such assessment can effectively show the capability preservation of the model before and after compression. Thus, the superior results of our method in Table 1 and Table 2 in the paper also show its favorable generalizability and cross-task portability. Following your suggestion, we also conduct experiments to use samples of LLaVA-Description for offline estimation and evaluate on MM-Vet to inspect the generalizability. For baseline, we also use the samples of MM-Vet for offline estimation. We present the PPL results based on LLaVA-1.5-7B in the below table. It further shows that our method can maintain favorable cross-task performance, exhibiting portability and generalizability.
>
> | Offline estimation | 10%  | 20%  | 30%  | 40%  | 50%  | 60%  | 70%  | 80%  | 90%  |
> | ------------------ | ---- | ---- | ---- | ---- | ---- | ---- | ---- | ---- | ---- |
> | LLaVA-Description  | 7.40 | 5.99 | 5.72 | 5.53 | 5.50 | 5.44 | 5.38 | 5.28 | 5.28 |
> | MM-Vet             | 7.38 | 5.97 | 5.66 | 5.53 | 5.50 | 5.41 | 5.38 | 5.28 | 5.28 |

---

> > ### Comment · Reviewer_hgee · 2025-08-06
> >
> > Thanks to the authors for their detailed responses. The rebuttal provides additional experiments to validate the method, which address my concerns. I would like to maintain my positive rating.

---

### Decision · Program_Chairs · 2025-09-17

**Decision:**

Accept (poster)

**Comment:**

This paper proposes PrefixKV, a framework for adaptive per-layer KV cache compression in large vision-language models, with a focus on efficient inference.

The paper is well-motivated, providing a principled analysis of inter-layer heterogeneity.  PrefixKV introduces an efficient binary search strategy to allocate compression ratios per layer under a global budget. The method achieves substantial inference speedups and state-of-the-art performance across multiple models and datasets.
Related approaches (e.g., PyramidKV) have been proposed previously, and the main advancement lies in the automated allocation approach rather than a new paradigm.

Recommendation:
Acceptance. The paper meets standards of soundness and originality for a conference paper, with reasons for acceptance outweighing reasons for rejection, although some missing clarity merit caution. Further discussion of limitations, clarifications, and more direct comparison with prior adaptive methods (e.g., PyramidKV) are encouraged for the camera-ready version.